# Controlling astrocyte-mediated synaptic pruning signals for schizophrenia drug repurposing with deep graph networks

**Alessio Gravina**[1]*, **Jennifer L. Wilson**[2], **Davide Bacciu**[1], **Kevin J. Grimes**[2], **Corrado Priami**[1]

**1** Department of Computer Science, University of Pisa, Pisa, Italy, **2** Department of Chemical & Systems Biology, Stanford University, Stanford, California, United States of America

* alessio.gravina@phd.unipi.it

**Data Availability Statement:** Data and Code are released on GitHub https://github.com/gravins/DGNs-for-schizophrenia.

## Abstract

Schizophrenia is a debilitating psychiatric disorder, leading to both physical and social morbidity. Worldwide 1% of the population is struggling with the disease, with 100,000 new cases annually only in the United States. Despite its importance, the goal of finding effective treatments for schizophrenia remains a challenging task, and previous work conducted expensive large-scale phenotypic screens. This work investigates the benefits of Machine Learning for graphs to optimize drug phenotypic screens and predict compounds that mitigate abnormal brain reduction induced by excessive glial phagocytic activity in schizophrenia subjects. Given a compound and its concentration as input, we propose a method that predicts a score associated with three possible compound effects, i.e., reduce, increase, or not influence phagocytosis. We leverage a high-throughput screening to prove experimentally that our method achieves good generalization capabilities. The screening involves 2218 compounds at five different concentrations. Then, we analyze the usability of our approach in a practical setting, i.e., prioritizing the selection of compounds in the SWEET-LEAD library. We provide a list of 64 compounds from the library that have the most potential clinical utility for glial phagocytosis mitigation. Lastly, we propose a novel approach to computationally validate their utility as possible therapies for schizophrenia.

## Author summary

Phagocytosis is a fundamental biological process to protect biological organisms from exogenous infectious particles as well as to preserve equilibrium and efficiency of the host by removing its unwanted cells. A dysregulation of the phagocytic activity can lead to severe consequences for the host. In this study, we focus on a recent theory that relates an excessive phagocytic activity in brain cells, and a consequent abnormal reduction in brain volume, to the development of schizophrenia. Our working hypothesis is that pharmaceutical compounds that can reduce excessive of phagocytic activity might prove effective as a schizophrenia treatment. Rather than attempting to develop ex-novo such a chemical compound, we rely on a more cost-effective and efficient approach that seeks candidate

**Funding:** The authors received no specific funding for this work.

**Competing interests:** The authors have declared that no competing interests exist.

therapies in a set of approved chemical compounds. To achieve this, we train a machine learning model capable of predicting, with good accuracy, the ability of a molecular compound to increase or decrease phagocytosis in the target brain cells. Our approach leverages learning models capable of directly processing the molecular graph of the compound, leading to the identification of 64 candidate drugs of potential clinical utility.

This is a *PLOS Computational Biology* Methods paper.

## Introduction

Schizophrenia is a chronic and severe mental disorder that affects how a person thinks, feels, and behaves. It is expressed as a combination of symptoms, such as recurrent psychosis, social withdrawal, anhedonia, and cognitive dysfunctions. Worldwide about 1% of the population is diagnosed with schizophrenia, with 100,000 new cases annually only in the United States [1].

A recent study [2] states that brain volumes, measured on Magnetic Resonance Imaging (MRI) scans, are abnormal in patients with schizophrenia compared to unaffected individuals, with a reduction in both grey and white matter. In particular, the decreased density of dendritic spines in schizophrenia subjects has been supposed by MacDonald et al. [3] as the result of an excessive pruning activity against synapses. This action is assumed to be performed by glial cells, which are non-neuronal cells with multiple functions in the central nervous system to support and remove neurons. This assumption is supported by the evidence that glial phagocytic activity may be directly associated with the prevalence of various neurodegenerative diseases due to hyperactivation of phagocytic pathways [4, 5]. In addition, the novel PET tracer binds to synaptic vesicle glycoprotein 2A (SV2A) and shows diminished uptake in the frontal and anterior cingulate cortex in individuals with schizophrenia [6].

Towards the goal of discovering novel treatments for schizophrenia, previous work conducted a large-scale phenotypic screen to discover compounds with the ability to alter glial cell phagocytosis. However, understanding structure activity relationships is a challenge in these screens. Further, generating accurate models is difficult because it is not known what chemical information is most associated with predicting chemical function and how to best represent this information for predictive models. Additionally, further experiments remain the gold standard for validating model predictions. Yet, it is not always possible to conduct additional high-throughput screens and we require alternative methods for testing the utility of model predictions.

The compound property/activity prediction problem is a task faced by pharmaceutical companies and academia to improve the comprehension of diseases, discover new drugs, or identify new indications of existing drugs. It is standard practice to scan large libraries of compounds to test their biological activity. However, this operation can be costly and time-consuming, and Machine Learning (ML) methods can be helpful to reduce the effort needed to run experiments. For that reason, in the last decades, several computational approaches have been proposed to determine compound properties, or as filter to select the most promising compounds for clinical and biological experiments [7, 8]. A pioneering work is that by Bianucci et al. [9], where the authors employed Cascade Correlation Networks for structures to predict the boiling point of Alkenes and to predict the affinity towards the Benzodiazepine/

GABA$_A$ receptor by a group of Benzodiazepines. More recently, Banerjee et al. [10] have developed a ML model to discriminate between sweet and bitter taste of molecules. Specifically, the model leverages a static fingerprint of the molecule to predict the property through a Random Forest. Similarly, Lind et al. [11] feed a static fingerprint and oncogene mutation status to a Random Forest to predict the activity versus inactivity of drugs against cancer cell lines. These results demonstrate that it is possible to associate chemical information to biological outcomes. Yet, static fingerprints are not sufficient in all applications.

Explicitly for schizophrenia, Zhao et al. [12] explored five different ML approaches to repurpose drugs for schizophrenia, depression, and anxiety disorders. In particular, they considered Deep Neural Networks, Support Vector Machines, Elastic Net regression, Random Forest, and Gradient Boosted Trees. Models were trained to predict whether a drug is a known treatment for the disease or not, using drug expression profiles as inputs. Those profiles capture transcriptomic changes when HL60, PC3, and MCF7 cell lines were treated with a chemical. Xu et al. [13] proposed PhenoPredict, a ranking algorithm for schizophrenia drug repurposing. PhenoPredict infers drug treatments from diseases that are phenotypically related to schizophrenia. These models demonstrate the ability to connect chemical information to biological information, yet they are limited to predicting molecular changes (such as gene expression) and are not suited to predicting phagocytic activity from phenotypic screens.

This work studies the use of ML techniques to predict the effects of compounds on glial phagocytic activity that cause abnormal brain reduction in schizophrenia subjects. This work has been done in collaboration with SPARK at Stanford University [14, 15], the main node of a partnership network between university and industry experts in chemistry, biology, and medicine to advance academic biomedical research discoveries into promising new treatments for patients. The objective of this work is to propose a ML method apt to optimize drug phenotypic screens. Specifically, our method identify compounds that reduce glial phagocytic activity for the treatment of schizophrenia, which, to the best of our knowledge, has not been proposed before. Our contribution can be summarized as follows. First, we introduce a ML method based on Deep Graph Networks to predict if a compound can influence the glial phagocytic activity in the brain tissue. Then, we evaluate our method on a real high-throughput screening experiment provided by SPARK. The proposed model achieves a macro Area Under the ROC curve (AUROC) of 0.68 when predicting if a compound inhibits, intensifies, or does not affect the phagocytic activity. Afterwards, we perform an analysis to understand the potential benefits of our approach in a practical scenario. Specifically, we leverage our method to prioritize the selection of a new set of compounds in the SWEETLEAD library, leading to the identification of 64 potential candidates. Lastly, we propose a novel approach to understand the relevance of compounds to biological use case. That approach allows us to compare our results with the more than 287,000 references in the literature. With this analysis we highlight the effectiveness of the model in identify compounds that are already studied in relation to brain-related diseases.

## Results and discussion

### Compound dissimilarity analysis

Before constructing the ML model, we measured the dissimilarity between compounds in the dataset to better understand how much molecular structures differ from each other in our data. A library of highly similar compounds would prevent the model from sufficiently discriminating drug effects. For this analysis, we computed the *Extended-Connectivity Fingerprint* (ECFP) for each compound, using standard parameters (i.e., radius equal to 3 and length equal to 1024); then, we measured the dissimilarity for each pair of fingerprints. Note that the ECFP

fingerprint is a non-adaptive vectorial representation that encodes the structure of a molecule (please refer to the Model section for a detailed description). For this analysis we leveraged two different metrics: *Cosine distance*, and *Jaccard dissimilarity*. Given *A* and *B* two vectors, the cosine distance is defined as

$$Cosine(A, B) = 1 - \frac{AB}{\| A \|_2 \| B \|_2}$$

while the Jaccard dissimilarity is

$$Jaccard(A, B) = 1 - \frac{|A \cap B|}{|A \cup B|}$$

as usual. Let's consider $\mathbf{d} \in \mathbb{R}^{n \times n}$ the squared upper triangular matrix where the element $\mathbf{d}_{i,j}$ represents the dissimilarity between fingerprints *i* and *j*, measured with one of the two metrics defined before. The final dissimilarity score is computed as

$$dissimilarity = \frac{1}{m} \sum_{i=0}^{n} \sum_{j>i}^{n} \mathbf{d}_{i,j}$$

where $m = n^2/2$.

Table 1 reports the final scores computed with the two metrics. Both scores are close to 1, highlighting that the compounds have distinct fingerprints, and therefore can be considered strongly dissimilar.

Afterward, we measured the scaffold diversity in our data with the same strategy proposed for the compound dissimilarity analysis. It is clear from Table 2 that there is a rich diversity even between scaffolds. Fig 1 emphasizes such dissimilarity by showing that only a few molecules in our data share the same scaffold. Indeed, there are 863 scaffolds associated with one molecule in the data, while only 26 are shared among 8 and 232 molecules. Such results highlight that the scaffold distribution is long-tailed. Lastly, Fig 2 shows through the Principal Component Analysis (PCA) that there is no evidence of any natural cluster neither in molecule or scaffold fingerprints. Thus, it is reasonable to assume that the compounds' uniqueness in the dataset represents an index of the strong complexity of the task.

## Model selection and risk assessment

The best model used in our experiments has been selected by an empirical analysis on the SPARK's high-throughput screening results (please refer to the Dataset section for details). We tested multiple models to understand which best predicted effects from our phenotypic screen. Each model used different information about the chemical structures in making predictions.

**Table 1. Cosine distance and Jaccard dissimilarity between dataset's compounds.**

| Name | Avg ± Std | 25 Perc. | 50 Perc. | 75 Perc. | Max | Min |
|------|-----------|----------|----------|----------|-----|-----|
| Cosine | 0.8189 ± 0.0281 | 0.7976 | 0.8162 | 0.8374 | 0.9476 | 0.7464 |
| Jaccard | 0.9026 ± 0.0173 | 0.8897 | 0.9017 | 0.9135 | 0.9877 | 0.8571 |

**Table 2. Cosine distance and Jaccard dissimilarity between dataset's compound scaffolds.**

| Name | Avg ± Std | 25 Perc. | 50 Perc. | 75 Perc. | Max | Min |
|------|-----------|----------|----------|----------|-----|-----|
| Cosine | 0.8234 ± 0.0750 | 0.7805 | 0.8092 | 0.8635 | 0.9999 | 0.7050 |
| Jaccard | 0.8990 ± 0.0341 | 0.8838 | 0.9033 | 0.9182 | 0.9772 | 0.8243 |

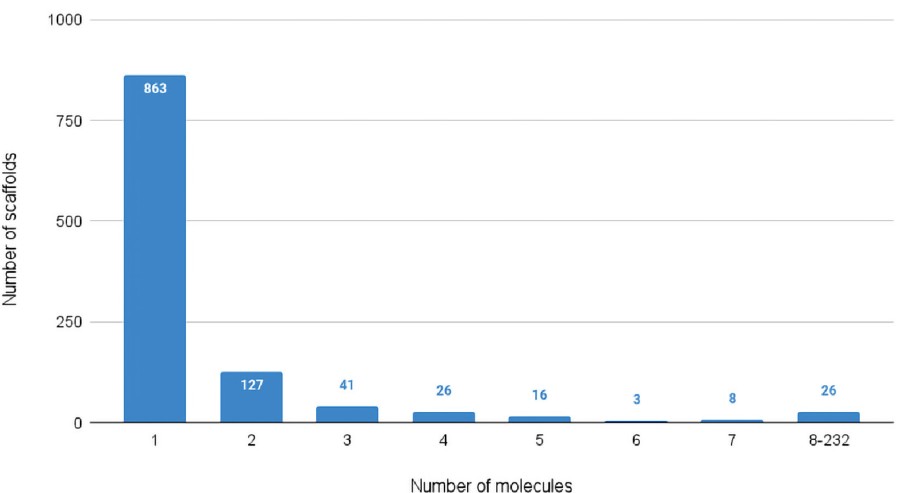

**Fig 1. The distribution of the number of shared scaffolds within molecules in the dataset.**

These models include: MoRF, MoNN, LinNN, SAGENN, GaNN, ENN, and NeFPNN. The first two exploit static compound fingerprints by leveraging the ECFP technique. LinNN leverages adapting fingerprints computed by a MLP using only atom information. The latter employ several deep graph network based fingerprints, which exploit atom and topological information, with ENN and NeFPNN including also bond features. Please refer to the Model section for details.

Table 3 reports the predictive performance achieved in the 3-fold cross-validation on the development set, using the standard stratification schema that divides the compounds maintain the distribution of the target variable. All configurations outperform the baseline LinNN, whose performance is very close to a random guesser. However, ENN and NeFPNN resulted in a validation score that is substantially inferior with respect to the best performing models. These results suggest that bond information is not helpful in solving this task. The top four models have overlapping validation scores, but highly different training scores. This situation hints at the use of alternative stratification strategies.

Given the neat performance discrepancy between models, we ran a more articulated stratification only for the top four configurations, i.e., MoNN, MoRF, SAGENN, GaNN. The

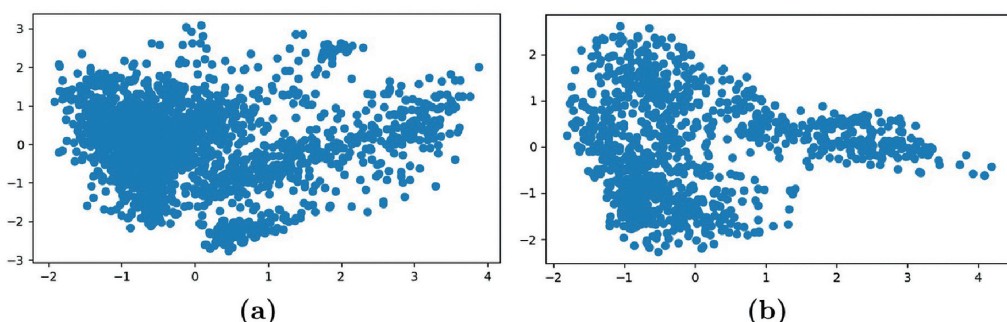

**Fig 2.** (a) The PCA plot of the molecule ECFPs. (b) The PCA plot of the scaffold ECFPs.

**Table 3. Macro-AUROC scores achieved with simple cross-validation.**

| Model | Train | Valid |
|---|---|---|
| MoNN | $0.9018_{\pm0.0192}$ | $0.6759_{\pm0.0020}$ |
| SAGENN | $0.8999_{\pm0.0134}$ | $0.6688_{\pm0.0118}$ |
| MoRF | $0.9361_{\pm0.0031}$ | $0.6674_{\pm0.0082}$ |
| GaNN | $0.8418_{\pm0.0052}$ | $0.6548_{\pm0.0148}$ |
| NeFPNN | $0.7102_{\pm0.0058}$ | $0.6334_{\pm0.0086}$ |
| ENN | $0.6840_{\pm0.0121}$ | $0.6021_{\pm0.0093}$ |
| LinNN | $0.5007_{\pm0.0008}$ | $0.5016_{\pm0.0011}$ |

**Table 4. Macro-AUROC scores achieved with complex cross-validation.**

| Model | Train | Valid |
|---|---|---|
| MoRF | $0.9062_{\pm0.0051}$ | $0.6764_{\pm0.0046}$ |
| GaNN | $0.8301_{\pm0.0068}$ | $0.6680_{\pm0.0039}$ |
| SAGENN | $0.8751_{\pm0.0111}$ | $0.6547_{\pm0.0031}$ |
| MoNN | $0.8580_{\pm0.0182}$ | $0.6520_{\pm0.0038}$ |

complex strategy separates compounds into groups by maintaining high compound diversity in each split of the data (details discussed in the Experimental setting section). Table 4 shows that nearly all models have better (or overlapping) validation performances with respect to the simple cross-validation strategy discussed above. Training scores are 1 to 5 points smaller than in Table 3 and the validation scores have overlapping value ranges, but the training scores have less optimistic values. MoRF and GaNN are the configurations with the highest gain in validation with the complex stratification setting, while MoNN shows a reduction of approximately 2 points.

After the model selection phase, where we selected the best hyper-parameters configuration for each model, we performed the risk assessment step by evaluating the models on the hold-out test set. Also in this case we considered only MoNN, MoRF, GaNN, and SAGENN. Table 5 shows that the configurations selected with the simple cross-validation often obtained a better performance than those selected with the complex stratification strategy. MoRF is the only model achieving an higher score in the latter case. With both methods, each model reached a performance in line with that obtained during model selection.

By analyzing the confusion matrices in Fig 3, we observe that a higher performance corresponds to an increased ability to correctly predict both *increase phagocytosis* and *decrease phagocytosis* classes. This result suggests that DGN-based models are more capable than the Morgan-based counterpart to recognize significant patterns to tackle this central task in our study.

**Table 5. Comparison of macro-AUROC scores computed during risk assessment.**

| Model | Simple | | Complex | |
|---|---|---|---|---|
| | Train | Test | Train | Test |
| GaNN | 0.8397 | 0.6877 | 0.8397 | 0.6632 |
| SAGENN | 0.8466 | 0.6767 | 0.8466 | 0.6679 |
| MoNN | 0.8991 | 0.6705 | 0.8991 | 0.6386 |
| MoRF | 0.8786 | 0.6676 | 0.8833 | 0.6701 |

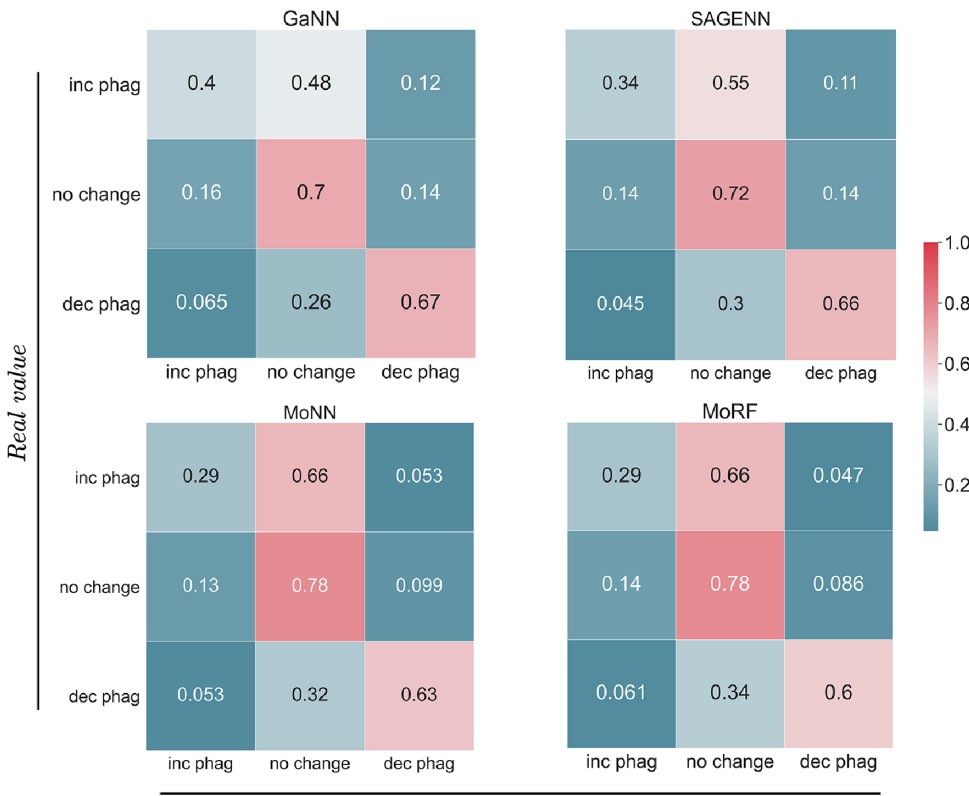

**Fig 3. Normalized confusion matrices of GaNN, SAGENN, MoNN, and MoRF on test set.**

Upon investigation of the confusion matrices, models have relatively high prediction accuracy for predicting drugs with no effect on phagocytosis (center square on all matrices). All models performed well in predicting drugs that would decrease phagocytosis (bottom right of each matrix) with the GaNN having the highest accuracy for this compound effect. All models had relatively low prediction accuracy for compounds that increase phagocytosis (upper left square of each matrix) and erred on the side of predicting no effect for these drugs (upper middle square of each matrix). For our dataset, we were prioritizing compounds that decreased phagocytosis, and so relatively high prediction for these compounds relative to the increase phagocytosis group was suitable. Ultimately, we selected the GaNN as the best model for both its performance on the hold-out test set and its capacity to distinguish between the three classes.

## Problem relaxation

To prove the complexity of the task, we relaxed the problem to predict whether a compound has a *positive* or *negative* impact on phagocytosis. We considered *no change* and *increase phagocytosis* classes as the negative label. Each model takes as input a compound and its concentration to predict the impact on phagocytosis. Thus, we shifted from multi-label to a binary classification problem.

We followed the same experimental procedure as in the Model selection and risk assessment section. We first performed a model selection phase by running a 3-fold cross-validation on the development set using the standard stratification schema. Given the strong discrepancy

**Table 6. Comparison of macro-AUROC scores achieved with simple and complex cross-validation when only 2 classes are considered.**

| Model | Simple | | Complex | |
|---|---|---|---|---|
| | Train | Valid | Train | Valid |
| SAGENN | $0.9313_{\pm 0.0015}$ | $0.7898_{\pm 0.0106}$ | $0.9090_{\pm 0.0004}$ | $0.7825_{\pm 0.0044}$ |
| MoNN | $0.9364_{\pm 0.0260}$ | $0.7791_{\pm 0.0150}$ | $0.9405_{\pm 0.0013}$ | $0.7895_{\pm 0.0013}$ |
| GaNN | $0.9094_{\pm 0.0076}$ | $0.7770_{\pm 0.0034}$ | $0.8995_{\pm 0.0137}$ | $0.7776_{\pm 0.0104}$ |
| MoRF | $0.9522_{\pm 0.0380}$ | $0.7702_{\pm 0.0170}$ | $0.9588_{\pm 0.0016}$ | $0.7731_{\pm 0.0049}$ |
| ENN | $0.9065_{\pm 0.0283}$ | $0.7662_{\pm 0.0035}$ | – | – |
| LinNN | $0.9212_{\pm 0.0088}$ | $0.7251_{\pm 0.0085}$ | – | – |
| NeFPNN | $0.8350_{\pm 0.0822}$ | $0.7205_{\pm 0.0191}$ | – | – |

**Table 7. Comparison of macro-AUROC scores computed during risk assessment when only 2 classes are considered.**

| Model | Simple | | Complex | |
|---|---|---|---|---|
| | Train | Test | Train | Test |
| SAGENN | 0.8687 | 0.7694 | 0.8684 | 0.7533 |
| MoNN | 0.9348 | 0.7676 | 0.9345 | 0.7634 |
| MoRF | 0.9262 | 0.7445 | 0.9372 | 0.7500 |
| GaNN | 0.8457 | 0.7368 | 0.8852 | 0.7376 |

between validation and training scores obtained in this phase, we selected the top four model configurations and we run the more articulated stratification. Table 6 reports the results of both experiments. As it shows, relaxing the problem to binary classification helps improve the final performances. Indeed, the validation scores with the standard stratification are more than 10 points higher on average if compared to multi-label classification. When the more sophisticated stratification schema is employed, the difference increases to over 11 points. Particularly interesting is the improvement of LinNN, which performance is on par with NeFPNN. This result remarks the fact that bond information is not strictly helpful in solving this task.

After the model selection, we proceeded with the risk assessment phase only for the most performing configurations. Table 7 shows the results obtained during this step. The scores exhibit similar behavior to the validation scores with respect to the original problem. However, they are below the confidence interval discovered in the model selection phase. We believe that this is the consequence of some labeling noise introduced during class aggregation. Indeed, the class *no change* is a borderline class that may contain noise itself. For such reasons, we believe that the three-class splitting leads to better performances that are less prone to errors.

These findings show that by relaxing the problem it is possible to achieve higher performances. Therefore, it is reasonable to assume these results as proof of the complexity of the original problem. Indeed, in this case, the computational task of structure-function prediction is really hard because phagocytosis is a multi-protein biological process. This suggests that multiple molecules could bind proteins in this pathway and that there might not be one ideal structure for influencing this process. Moreover, this situation, followed by the high dissimilarity between molecule structures, indicates that our method learns good compound representations and is not overfitting molecular sub-structures for making predictions.

## SWEETLEAD library repurposing

The main goal of our analysis is understanding the usability of our approach in a practical setting, i.e., prioritizing the selection of compounds to be tested in a biomedical experiment. We

**Table 8. The number of predicted compounds in SWEETLEAD library for each class.**

|  | FDA approved | Complete library |
|---|---|---|
| no change | 585 | 1647 |
| decrease phagocytosis | 309 | 964 |
| increase phagocytosis | 143 | 431 |
| mixed* | 354 | 1272 |

\* model's outcome change with different dosage.

focus on the GaNN model which has emerged as the best configuration according to our model selection and risk assessment analysis. We leveraged the model to predict the impact, at different dosages, of a new set of compounds on astrocyte-mediated synaptic pruning in schizophrenia. We considered the compounds in the SWEETLEAD library [16], an in silico database of approved drugs, regulated chemicals, and herbals designed for drug discovery. The library contains 4314 compounds with 1391 of them marked as FDA approved.

We simulated with the GaNN model the impact of each compound in SWEETLEAD on the phagocytic activity at five different concentrations, i.e., 1.39, 2.78, 5.56, 11.11, 22.22 $\mu$M, to parallel the dose ranges used in the initial phenotypic screen and to mirror those typically used in literature.

Table 8 shows how many compounds in SWEETLEAD are predicted to belong to each of the three phagocytosis classes by our GaNN model. The GaNN generates different outcomes for different doses only in 29.49% of the compounds, so the model can be considered highly confident on the other predictions.

We analyzed predicted compound effects by Anatomical Therapeutic Chemical (ATC) codes to understand the therapeutic use classification of these drugs. Additionally, ATC codes related to the neurological system would provide further evidence that a predicted drug could modify phagocytosis in the brain. ATC codes are part of a classification system, controlled by World Health Organization, that classifies drugs according to the organ or system on which they act and their therapeutic, pharmacological and chemical properties. ATC codes are hierarchically organized in five levels, where the first one is the more general and refers to the anatomical main group, while the latter are the more specific and indicates the chemical substance. Note that a drug can be associated with more than one ATC code.

We focused on FDA-approved compounds that have a match in DrugBank [17]. Fig 4 shows the first level of ATC codes with respect to each class. The *decrease phagocytosis* class is mostly characterized by the category **N** (nervous systems), confirming that the model can identify compounds that are already used to treat nervous system diseases. Intriguinly, the next highest category for *decrease phagocitosys* class is **C** (cardiovascular system). There is some literature evidence supporting the repurposing of cardiovascular system drugs for neurological conditions. Specifically, beta-blockers can reduce severity of migraines, and statins can reduce contrast-induced neuropathy [18]. This preliminary evidence suggests that the model may be predicting viable applications of these drugs to affect phagocytosis. The Cardiovascular system class drugs also contained the highest number of predicted increase phagocytosis drugs suggesting that we cannot predict new drug effects from ATC codes alone, and that even within-class drugs can have distinct effects.

We selected 64 compounds that have most potential clinical utility. The selection was among compounds marked as FDA approved and that have been predicted to decrease phagocytosis with high confidence. The complete list is reported in Table 9.

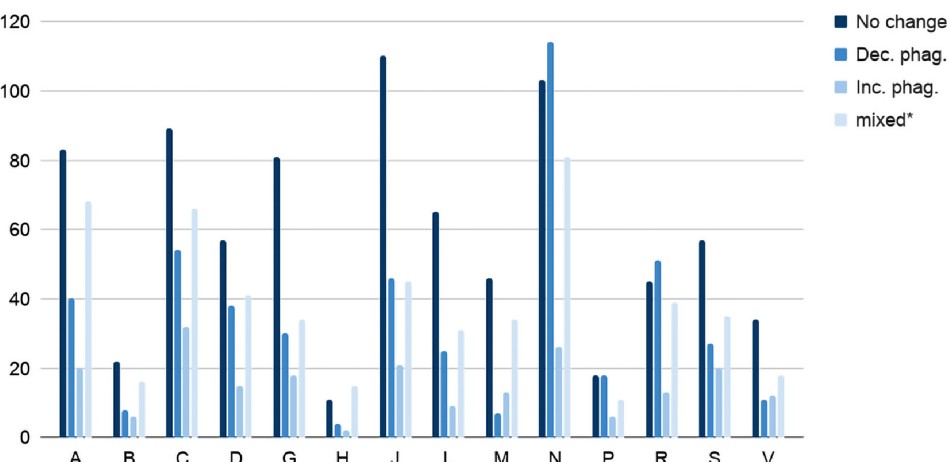

**Fig 4. First level ATC codes matched with each predicted class.** The multiple vertical bars chart the first level ATC codes with respect to the predicted class for each analysed compound. The letters on the x-axis refer to the ATC codes; A = alimentary tract and metabolism, B = blood and blood forming organs, C = cardiovascular system, D = dermatologicals, G = genito-urinary system and sex hormones, H = systemic hormonal preparations, excluding sex hormones and insulins, J = antiinfectives for systemic use, L = antineoplastic and immunomodulating agents, M = musculo-skeletal system, N = nervous system, P = antiparasitic products, insecticides and repellents, R = respiratory system, S = sensory organs, and V = various. We used the label *mixed** to indicate those cases where the model prediction changes with different doses. We recall also that each compound may be associated with multiple ATC codes.

We sought to computationally validate the utility of the selected compounds as possible therapies for schizophrenia. We exploited SciFinder [19], which is a database for chemical literature, to extract the 15 topics with highest frequencies per compound. We hypothesized that biological terms co-mentioned with drug names in the literature could represent a reasonable measure of the goodness of candidates. We recall that a topic is a term used by the Chemical Abstracts database to identify the general topic of a reference. We conducted the analysis over more than 287,000 references. Chemical compounds are associated with an average of 914 terms, and frequency of chemical-term co-mentions are in the tens to hundreds of mentions. We retain the top 15 topics with highest frequencies for further analysis. For our 64-chemicals, the frequencies for the 15 topics ranged from 6992 to 1. This level of co-mention was sufficient for us to analyze whether these compounds had utility for our repurposing goal.

We clustered the compounds into three main groups based on topics:

- **Brain** ($\mathcal{B}$), which contains all compounds with at least a brain-related topic in the top-15. We considered as brain-related topics the terms: Antidepressants, Antipsychotics, Bipolar disorder, Brain, Cognitive disorders, Depression, Epilepsy, Mental and behavioral disorders, Mood disorder, Multiple sclerosis, Obsessive-compulsive disorder, Parkinson disease, Psychosis, Schizophrenia;

- **Antibiotics** ($\mathcal{A}$), which contains all compounds with the topic Antibiotic in the top-15;

- **Miscellaneous** ($\mathcal{M}$), which contains all compounds with no brain or antibiotic related topics in the top-15.

We selected brain-related topics because this could suggest applications to neurological diseases. We also selected antibiotics because these compounds are generally well-tolerated and have known safety profiles. Moreover, some literature evidence suggest that antibiotics can

**Table 9. The list of selected compounds ranked by model confidence.**

|  | DrugBank ID | Name | Cluster |  | DrugBank ID | Name | Cluster |
|---|---|---|---|---|---|---|---|
| 1 | DB00822 | Disulfiram | $\mathcal{B}$ | 33 | DB00485 | Dicloxacillin | $\mathcal{A}$ |
| 2 | DB01163 | Amdinocillin | $\mathcal{A}$ | 34 | DB01061 | Azlocillin | $\mathcal{A}$ |
| 3 | DB00739 | Hetacillin | $\mathcal{A}$ | 35 | DB00377 | Palonosetron | $\mathcal{M}$ |
| 4 | DB01000 | Cyclacillin | $\mathcal{A}$ | 36 | DB00283 | Clemastine | $\mathcal{A}$ |
| 5 | DB00937 | Diethylpropion | $\mathcal{M}$ | 37 | DB00481 | Raloxifene | $\mathcal{M}$ |
| 6 | DB00363 | Clozapine | $\mathcal{B}$ | 38 | DB00713 | Oxacillin | $\mathcal{A}$ |
| 7 | DB00408 | Loxapine | $\mathcal{B}$ | 39 | DB00883 | Isosorbide dinitrate | $\mathcal{M}$ |
| 8 | DB06209 | Prasugrel | $\mathcal{M}$ | 40 | DB00319 | Piperacillin | $\mathcal{A}$ |
| 9 | DB00334 | Olanzapine | $\mathcal{B}$ | 41 | DB01078 | Deslanoside | $\mathcal{B}$ |
| 10 | DB00462 | Methscopolamine bromide | $\mathcal{B}$ | 42 | DB00208 | Ticlopidine | $\mathcal{M}$ |
| 11 | DB05271 | Rotigotine | $\mathcal{B}$ | 43 | DB00572 | Atropine | $\mathcal{B}$ |
| 12 | DB00354 | Buclizine | $\mathcal{B}$ | 44 | DB01147 | Cloxacillin | $\mathcal{A}$ |
| 13 | DB01224 | Quetiapine | $\mathcal{B}$ | 45 | DB01104 | Sertraline | $\mathcal{B}$ |
| 14 | DB13996 | Magnesium acetate | $\mathcal{B}$ | 46 | DB01221 | Ketamine | $\mathcal{B}$ |
| 15 | DB00514 | Dextromethorphan | $\mathcal{M}$ | 47 | DB00219 | Oxyphenonium | $\mathcal{B}$ |
| 16 | DB00857 | Terbinafine | $\mathcal{A}$ | 48 | DB06787 | Hexocyclium | $\mathcal{M}$ |
| 17 | DB00245 | Benzatropine | $\mathcal{B}$ | 49 | DB00938 | Salmeterol | $\mathcal{M}$ |
| 18 | DB06718 | Stanozolol | $\mathcal{M}$ | 50 | DB00539 | Toremifene | $\mathcal{M}$ |
| 19 | DB00882 | Clomifene | $\mathcal{M}$ | 51 | DB00831 | Trifluoperazine | $\mathcal{B}$ |
| 20 | DB00758 | Clopidogrel | $\mathcal{M}$ | 52 | DB01160 | Dinoprost tromethamine | $\mathcal{M}$ |
| 21 | DB00561 | Doxapram | $\mathcal{M}$ | 53 | DB00341 | Cetirizine | $\mathcal{M}$ |
| 22 | DB00496 | Darifenacin | $\mathcal{B}$ | 54 | DB00390 | Digoxin | $\mathcal{M}$ |
| 23 | DB06230 | Nalmefene | $\mathcal{M}$ | 55 | DB00419 | Miglustat | $\mathcal{M}$ |
| 24 | DB00392 | Profenamine | $\mathcal{B}$ | 56 | DB01328 | Cefonicid | $\mathcal{A}$ |
| 25 | DB00925 | Phenoxybenzamine | $\mathcal{B}$ | 57 | DB00474 | Methohexital | $\mathcal{B}$ |
| 26 | DB00679 | Thioridazine | $\mathcal{B}$ | 58 | DB09346 | Metrizoic acid | $\mathcal{B}$ |
| 27 | DB00920 | Ketotifen | $\mathcal{A}$ | 59 | DB09363 | Rauwolfia serpentina root | $\mathcal{M}$ |
| 28 | DB01186 | Pergolide | $\mathcal{B}$ | 60 | DB06335 | Saxagliptin | $\mathcal{M}$ |
| 29 | DB00543 | Amoxapine | $\mathcal{B}$ | 61 | DB00355 | Aztreonam | $\mathcal{A}$ |
| 30 | DB01238 | Aripiprazole | $\mathcal{B}$ | 62 | DB00088 | Alglucerase | $\mathcal{M}$ |
| 31 | DB00865 | Benzphetamine | $\mathcal{B}$ | 63 | DB01180 | Rescinnamine | $\mathcal{B}$ |
| 32 | DB04844 | Tetrabenazine | $\mathcal{B}$ | 64 | DB06282 | Levocetirizine | $\mathcal{M}$ |

directly affect schizophrenia [20] and there is mounting evidence that the gut microbiome affects brain health [21, 22]. Lastly, we kept a miscellaneous category to capture drugs without a clear alignment to either of these hypotheses.

We report the associations between compounds and clusters in Table 9. It appears that compounds with brain-related topics have a higher ranked probability predicted by the GaNN model with respect to others. Indeed, 20 of the first 30 compounds belong to the $\mathcal{B}$ cluster. These additional results reinforce the idea that the model is effective in identifying compounds that are already studied in relation to brain-related diseases. Three of the top five compounds ranked by the GaNN model are antibiotics suggesting that repurposing for schizophrenia could leverage the human gut microbiome-brain connection.

Lastly, we found evidence of the utility of some of our 64 candidates (i.e., Loxapine, Dextro-methorphan, Thioridazine, Trifluoperazine, and Cetirizine) in the work of So et al. [23]. In their analysis, they leveraged GWAS data, and gene imputation to identify drug repurposing

candidates for multiple psychiatric diseases. Their work presents a complimentary view that our predicted candidates influence biology relevant to psychiatric disease. We observe that a high recall score is not possible in this case because the authors report only the top-15 compounds.

## Materials and methods

### Dataset

We performed our experiments on top of SPARK's high-throughput screening results, which aim was to screen for compounds that inhibit or activate MEGF10 [24] to correct aberrant astrocyte-mediated synaptic pruning in schizophrenia. With that purpose, the screen was a phagocytosis assay using astrocytes isolated from fetal human brain samples and synaptosomes prepared from mouse brain samples. They measured phagocytosis with a pH-sensitive fluorescent dye conjugated to the synaptosomes that is only activated when engulfed and localized to the low pH found in intracellular lysosomes. The screen was conducted in plates containing both positive and negative controls for data normalization (article in preparation).

The screening assessed 2218 different compounds at different concentrations, i.e., 1.39, 2.78, 5.56, 11.11, 22.22 $\mu$M. All the analyzed compounds derive from the *Library of Pharmacologically Active Compounds* (LOPAC) [25] and *NIH Clinical Collection* (NIHCC) [26], so they include inhibitors, receptor ligands, pharma-developed tools, and approved drugs. Due to the overlap of the two libraries, we removed duplicate results, leading to 10914 unique compound-dose combinations. For this analysis, we used median-normalized fluorescent signal as a proxy for relative phagocytosis.

We classified screening results by assigning a class to each instance. In particular, there are three class types:

- *increase phagocytosis*—if the combination of compound and dose *intensifies* the phagocytic activity;

- *decrease phagocytosis*—if the combination of compound and dose *inhibits* the phagocytic activity;

- *no change*—if the combination of compound and dose *does not affect* the phagocytic activity.

We assigned the classes by applying a threshold on the number of cells with phagocytosis signal. More specifically,

$$\begin{cases} \text{increase phagocytosis} & \text{if } \beta \geq 130\% \text{ of the controls} \\ \text{no change} & \text{if } 70\% < \beta < 130\% \text{ of the controls} \\ \text{decrease phagocytosis} & \text{if } \beta \leq 70\% \text{ of the controls} \end{cases}$$

with $\beta$ as the number of cells with phagocytosis signal with respect to the median of the plate.

Each compound is initially represented through its *Simplified Molecular Input Line Entry System* (SMILE) string. The SMILE string is a character string that captures the compounds' elements and the bonds between them. For each compound we also include additional information regarding atoms and bonds, as shown in Table 10.

Finally, given $\mathcal{D} = \{1.39, 2.78, 5.56, 11.11, 22.22\}$ the set of tested concentrations and $\mathcal{Y} = \{\text{increase phagocytosis}, \text{no change}, \text{decrease phagocytosis}\}$ the set of target values, the final dataset can be described as $\mathbb{D} = \{c_i, d_i, y_i\}_{i=1}^{10914}$, where $c_i$ is the $i$-th tested instance (note

**Table 10. The list of features regarding atoms and bonds.**

| Name | Type | Number of Values |
|------|------|------------------|
| Valence | one-hot | 7 |
| Number of *Hs* | one-hot | 6 |
| Hybridization | one-hot | 5 |
| Symbol | one-hot | 11 |
| Degree | one-hot | 7 |
| Aromatic | boolean | 2 |

**(a)** Atom features

| Name | Type |
|------|------|
| Aromatic | boolean |
| Single | boolean |
| Double | boolean |
| Triple | boolean |
| IsInRing | boolean |
| Conjugated | boolean |

**(b)** Bond features

that a single compound is tested under different concentrations), $d_i \in \mathcal{D}$ is the tested concentration for the instance, and $y_i \in \mathcal{Y}$ is the corresponding outcome.

For the purposes of our work, we represent each compound (originally expressed as a SMILE string) as a molecular graph, which is a consolidated representation of atoms and the bonds between them. We consider a molecular graph as an undirected graph defined as the tuple $g = (\mathcal{V}, \mathcal{E}, \mathbf{X}, \mathbf{E})$. The set $\mathcal{V}$ contains interacting entities, which in this case corresponds to atoms, and $\mathcal{E}$ is the set that contains the links among those entities, i.e., chemical bonds. On the other hand, $\mathbf{X} \in \mathbb{R}^{|\mathcal{V}| \times |F|}$ is the atom features matrix, where $|F|$ is the number of available features for an atom. Analogously, $\mathbf{E} \in \mathbb{R}^{|\mathcal{E}| \times |E|}$ is the bond features matrix, where $|E|$ is the number of available chemical bond features. We refer to $\mathbf{x}_i$ and $\mathbf{e}_{ij}$ as the feature vector of atom $i$ and feature vector of bond connecting atoms $i$ and $j$, respectively. Also, we denote the neighborhood of a node $i \in \mathcal{V}$ as the set, $\mathcal{N}(i) = \{j \in \mathcal{V} \,|\, \{i, j\} \in \mathcal{E}\}$, that contains every nodes with a link with node $i$.

## Model

We tackle the task of predicting glial phagocytic activity in brain tissue by an approach based on deep learning for graphs [27], given its power in encoding graph-based structures, e.g., molecules. Our model comprises the following two components:

- a compound embedding module $f_{comp}$;

- an output module $f_{out}$.

From a high level point of view, given $\mathcal{D}$ the set of tested concentrations, a compound, $c$, is processed as follows. First the compound is processed by the compound embedding module to compute a representation vector from its molecular graph,

$$\mathbf{h}_c = f_{comp}(c)$$

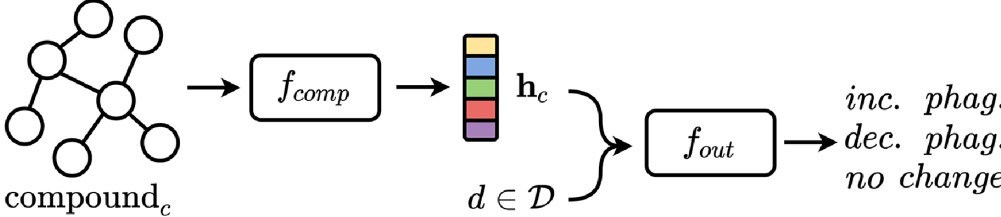

**Fig 5. A high level overview of the proposed model.** Given the molecular graph of a compound $c$ and a dose $d \in \mathcal{D}$, the model computes a vectorial representation $\mathbf{h}_c$ using the compound embedding module $f_{comp}$ and later computes the final result using the output module $f_{out}$ by leveraging the concatenation of $\mathbf{h}_c$ and $d$.

that later is concatenated with the dose, $d \in \mathcal{D}$, and passed to the output module. Hence, the final prediction is computed as

$$o = f_{out}(\mathbf{h}_c, d).$$

This process is summarized visually in Fig 5.

We considered three different compound embedding modules and two output modules and we empirically confronted their effectiveness. We implemented the output module either by a Multi-Layer Perceptron (MLP) or by a Random Forest (RF) [28]. For the compound embedding module, we first leveraged the *Extended-Connectivity Fingerprint* (ECFP) [29] based on the *Morgan algorithm* [30]. ECFPs are static molecular fingerprints that exploit atom neighborhoods to represent molecules through a non-adaptive approach that does not take into consideration the predictive task at hand. A visual representation of this process is shown in Fig 6.

Differently from the ECFP method, which is static, the other two approaches considered for the compound embedding module are adaptive and generate compound embeddings that are specialized for the specific predictive problem. To this end, we consider deep learning solutions that can process the compound represented as a molecular graph.

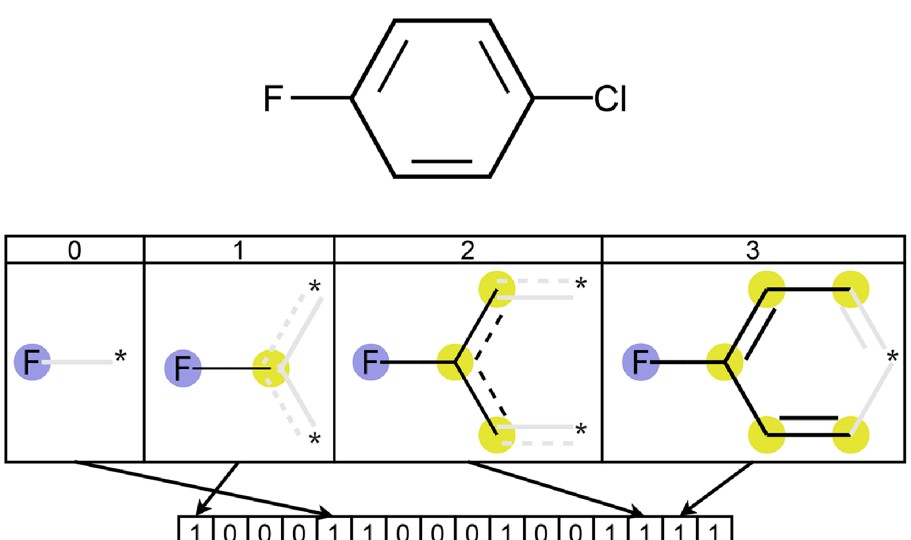

**Fig 6. Fluorine's influence in a circular fingerprint computation (radius equal to 3) of 1-Chloro-4-fluorobenzene.**

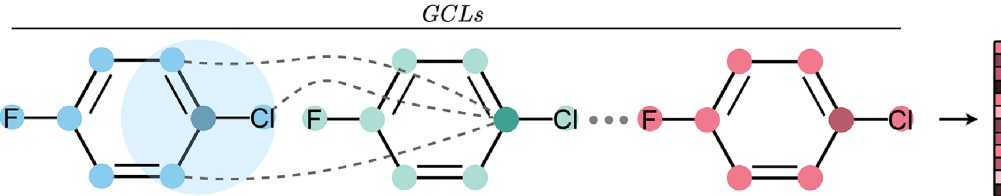

**Fig 7. An example of DGN applied on 1-Chloro-4-fluorobenzene molecular graph.** Given the molecular graph of a compound, each GCL computes the representation of each node in the graph as a transformation of their neighbor representations. In the end, node representations are aggregated to obtain a vectorial representation that reflect the original molecular graph.

The first adaptive approach that we consider is a linear embedding model, implemented through a MLP that computes a vectorial representation for each atom in the compound, which then are aggregated by an element-wise sum or mean without taking into consideration any spatial information. The second approach leverages Deep Graph Networks (DGNs) [27], which learn a mapping function that compresses the complex relational information captured by a graph into an information-rich feature vector that reflects both the topological and the label information in the original graph. At a high level, given a molecular graph as input, a DGN computes a representation for each node through transformations that combine the previous representation of the node with its neighbor representations. Those transformations are often referred to as Graph Convolutional Layers (GCLs). Ultimately, node representations are aggregated to obtain a single embedding for the whole graph. An overview of this method is shown in Fig 7.

We investigated different DGN implementations. More specifically, the DGNs that we consider are based on GraphSAGE [31], Graph Attention Network (GAT) [32], Edge-Conditioned Convolution Network (ECC) [33, 34], and Neural Graph Fingerprint [35]. An overview of such methods is reported in S1 Appendix.

For our purposes, we evaluated the performance of seven configurations of compound embedding module and output module:

- Linear Embedding and Neural Network (LinNN)—used as a baseline for this task;

- ECFP fingerprint based on Morgan algorithm and Random Forest (MoRF)—commonly used in literature for biomedical-based problems [10, 11, 36, 37];

- ECFP fingerprint based on Morgan algorithm and Neural Network (MoNN);

- GraphSAGE and Neural Network (SAGENN);

- GAT Network and Neural Network (GaNN);

- ECC Network and Neural Network (ENN);

- Neural Graph Fingerprint and Neural Network (NeFPNN).

### Experimental setting

We split the data into a development set (80%) for model selection and a test set (20%) for risk assessment. We consider the obtained test set as a hold-out, in other words, as a set of examples only used to estimate the generalization performance of the model, and never used during the

training phase. Internally to the development set, we used a 3-fold cross-validation for model selection. We generated each split in a stratified fashion. In other words, each split maintains the distribution with respect to a target variable. Specifically, we implemented stratification according to two strategies. The former splits the data by maintaining the distribution with respect to the target $y$; while the latter with respect to the target $y$, the cardinality of atoms in the compounds, and the concentrations. The rationale behind the latter strategy is to generate more homogeneous data splits, avoiding unbalanced distributions of molecule size and concentrations in the training and validation splits. We will refer to the first strategy as simple cross-validation, while the second as complex cross-validation.

The three classes are not balanced, with the *no change* category representing roughly 81% of samples. To preserve the minority classes we undersampled the majority class. At each epoch we randomly sampled from the *no change* instances to generate a subset with the same size of other classes. The rationale behind this approach is to keep the classes always balanced, and also to leverage all the data available. Indeed different sub-samples are extracted at each epoch. We recall that Random Forests are more resistant to data imbalance, therefore, we did not implement undersampling for RF-based models.

Our experiments can be summarized as follows. At first, we conduct the model selection and the risk assessment phases. The rational behind these steps is, first, to select the best hyper-parameters for the models among a set of candidates, and then to evaluate their generalization capability on a different set of data. Lastly, we use the best model, selected from the previous stage, in a real world drug repurposing scenario with the aim of understanding the potential in prioritizing the selection of compounds to be tested in a specific biomedical experiment.

We performed hyper-parameter tuning via grid search, optimizing the AUROC with macro-average, which is a good estimate of the classification performances since the dataset is balanced. We recall that the macro-AUROC is defined as

$$\text{macro} - \text{AUROC} = \frac{1}{c}\sum_{i=1}^{c}\text{AUROC}_i$$

where $c$ is the number of classes (in this work $c = 3$) and $\text{AUROC}_i$ is the metric computed for class $i$. The grids used in our experiments are reported in S2 Appendix. We trained models with $f_{out}$ = MLP to minimize the Cross-Entropy loss accumulated across all the instances in the dataset.

The experiments were carried on a Dell server with 4 Nvidia GPUs Tesla P100. We release openly the code implementing our methodology and reproducing our empirical analysis at: https://github.com/gravins/DGNs-for-schizophrenia.

## Conclusion

This work investigates the benefits of ML for graphs to predict compounds that mitigate abnormal brain reduction induced by excessive glial phagocytic activity in people affected by schizophrenia. In this context, we designed a model that is able to recognize whether a compound can reduce, increase, or not influence phagocytosis. More precisely, the model takes as input a compound and a concentration to predict a score associated with the three possible compound effects. This allows us to anticipate compounds with potentially desirable clinical effects for patients with schizophrenia. Internally, the model leverages a static fingerprint (i.e., Morgan-based ECFP) or an adapting fingerprint (i.e., DGNs) to represent compounds. We have shown experimentally that our approach is effective and has good generalization capabilities. Indeed, we have found that the model can generalize its predictions when employed on an

unseen library, identifying as potential beneficial compounds those already used to treat brain-related diseases. Lastly, we have presented a list of compounds that we believe have the most potential clinical utility against glial-mediated brain reduction in schizophrenia patients.

We tested multiple chemical representations and discovered that an adapting approach was sufficient for describing phenotypic screen effects. A static fingerprint was insufficient, yet including full bond information decreased model performance. This suggests that, in some cases, structure-function information requires knowledge of the atoms and their arrangements, but not the full detail of their connections. It may be advantageous for other rapid drug development programs to leverage this information to more efficiently predict compounds with desired effects. In this scenario, we were eager to test our model on new drug information and validate the utility of the model, however experimental validation was infeasible for this work. Instead, we tested a novel validation approach by using SciFinder to understand the relevance of compounds to biological use cases. Indeed, SciFinder recovered drug-phenotype associations that mapped to neurological terms, suggesting that emerging literature evidence supported the potential utility of model-predicted compounds. We anticipate that this can be used to further prioritize predicted compounds and minimize the needs for very large, follow-up validation screens and that other structure-functional studies using machine learning will benefit from our in silico validation approach.

## Supporting information

**S1 Appendix. Overview of the employed dynamic compound embedding modules based on DGNs.**
(PDF)

**S2 Appendix. Hyper-parameters tables.**
(PDF)

## Acknowledgments

This work has been partially supported by SPARK at Stanford University. The authors would like to thank SPARK members and Francesco Landolfi, University of Pisa, for the insightful discussions throughout the development of this work.

## Author Contributions

**Conceptualization:** Alessio Gravina, Jennifer L. Wilson, Davide Bacciu, Kevin J. Grimes, Corrado Priami.

**Formal analysis:** Alessio Gravina.

**Investigation:** Alessio Gravina.

**Methodology:** Alessio Gravina.

**Resources:** Jennifer L. Wilson, Kevin J. Grimes.

**Software:** Alessio Gravina.

**Supervision:** Jennifer L. Wilson, Davide Bacciu, Kevin J. Grimes, Corrado Priami.

**Validation:** Alessio Gravina.

**Writing – original draft:** Alessio Gravina, Jennifer L. Wilson, Davide Bacciu, Kevin J. Grimes, Corrado Priami.

**Writing – review & editing:** Alessio Gravina, Jennifer L. Wilson, Davide Bacciu, Kevin J. Grimes, Corrado Priami.

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
