## [Decision Letter · Decision Letter 0]

6 Jan 2022

Dear Mr. Gravina,

Thank you very much for submitting your manuscript "Controlling astrocyte-mediated synaptic pruning signals for schizophrenia drug repurposing with Deep Graph Networks" for consideration at PLOS Computational Biology.

As with all papers reviewed by the journal, your manuscript was reviewed by members of the editorial board and by several independent reviewers. In light of the reviews (below this email), we would like to invite the resubmission of a significantly-revised version that takes into account the reviewers' comments.

As you will see from the reviews, both reviewers criticize the fact that the code has not been made available in the GitHub repository, which did not allow an in depth assessment of the method. I therefore urge you to make the code available upon resubmission. You will also see that Reviewer 2 has concerns regarding the performance of the method, as well as on the strength on the validation results. From the methods point of view, the CV procedure should be also adapted according to the reviewer's comments. Reviewer 1 suggests a more concise description of the methods (e.g. in the "Compound dissimilarity analysis" section). Finally, the novelty with respect to already published LBVS methods should be stressed in your revised version.

We cannot make any decision about publication until we have seen the revised manuscript and your response to the reviewers' comments. Your revised manuscript is also likely to be sent to reviewers for further evaluation.

Sincerely,

Carl Herrmann, Ph.D.

Associate Editor

PLOS Computational Biology

Feilim Mac Gabhann

Editor-in-Chief

PLOS Computational Biology

Reviewer's Responses to Questions

**Comments to the Authors:**

Reviewer #1: This work presents a machine learning-based model that predicts the effects of the given compound and its concentration. Then, the authors screen a great number of compounds by using the proposed computational methods, and validate selected compounds as possible therapies for schizophrenia. I have the following concerns.

(1) The authors developed a computational method for schizophrenia drug repurposing with Deep Graph Networks. The authors directly employed Deep Graph Networks to their problems but didn’t propose novel methods.

(2) Do all compounds have the five concentrations, i.e., 1.39, 2.78, 5.56, 11.11, 22.22. If so, how can we make prediction if the compound has concentrations different from these?

(3) The authors introduce too much basic knowledge about the Deep Graph Networks and molecular fingerprint ECFP, and it is better to make them concise. The writing and organization could be improved for better readability.

(4)The source codes and datasets are not publicly available, and the web link provided by the author is empty. It is difficult to objectively evaluate the quality of the data and explore details about the model implementation.

Reviewer #2: “Controlling astrocyte-mediated synaptic pruning signals for schizophrenia drug repurposing with Deep Graph Networks.”

The authors present results of various ligand-based virtual screening models for predicting a phenotypic endpoint potentially related to schizophrenia. The models are trained and validated using data from SPARK HTS involving a cell-based assay of astrocytes with readout based on phagocytotic activity. Here 2218 diverse compounds from LOPAC and NIHCC were screened at 5 doses (10914 unique compound-dose combinations with activity classes). The best performing model model based on macro-AUROC on held-out subset, a GNN, was then applied on the SWEETLEAD compound in a retrospective test to examine model performance. Model performance here was examined by the extent to it prioritized known CNS active molecules as determined based on ATC labels, which associate the molecules with their likely tissue/organ target sites of action. The authors prioritize 64 molecules from SWEETLEAD based on predicted reduction in synaptic phagocytotic activity and check the likely targets of these compounds based on analysis of SciFinder chemical abstracts terms categorizing the molecules.

The development and testing of deep graph networks for ligand-based virtual screening on phenotypic endpoints is currently of high interest to the computational drug discovery field and PLOS Comp Biol readership. The integration of dose with the compound structure input representation is, to my knowledge, novel and potentially an advance.

The manuscript is clearly written and concise.

The codes/data used in the work are not yet available on the github repository indicated in the text.

Major Issues

The AUROC of the top models (after sweeping hyperparameters) in validation on the held-out SPARK HTS data seems low (0.65-0.68), regardless of cross-validation approach and also poor in the risk-assessment test (Table 4—AUROC 0.64-0.69), compared to published performances of other LBVS models. The complex CV procedure, with diversification of compounds within the folds, seems like a sound approach but did not meaningfully enhance model performances in validation, risk-assessment phases. Although the SPARK molecule set was examined in terms of structural diversity (based on Jaccard and Cosine distances of chemical fingerprints), it is possible that molecules of similar scaffold/chemotype were in spread across folds. This might be mitigated by clustering the training molecules and confining cluster IDs to specific folds. This ensures that predictions are made on chemotypes outside of the training set. Also, rather than showing the quartiles in Table 1, a plot of the distributions (as histograms) would be more informative—as well as some examples of molecule pairs at the different dissimilarity values. It is not obvious what the dissimilarity scores reflect—perhaps some statistics on scaffold diversity (Bemis-Murcko) would be useful to reader.

The instances from the SPARK dataset were defined as unique compound-dose pairs. The inclusion of dose as part of the input representation could be a significant advance—especially when predicting complex phenotypic endpointsg. However, given the poor model performances, and the typical variability of single concentration points in HTS assays, the reviewer recommends a using just the compound representations as inputs and assigning activity labels based on the full dose-response curves (potency or binary/tertiary classes). This will reduce the number of instances from ~10000 to ~2000 but this would likely improve reliability of the labels and simplify the relationship between inputs and outputs. It’s understood that some molecules showed both pro- and anti-phagocytotic activities depending on dose. Such complex responses exhibited by the compound instances could be a result of artifacts. Assay responses at the higher concentrations are probably less relevant and might merely reflect toxicity. Perhaps using the observed response (pro- or anti-phagocytotic) at the lowest concentration point could resolve this issue. Also, since decreased phagocytotic activity is the therapeutically relevant endpoint, perhaps a simple binary classification could be used based on achieving some threshold decrease in phagocytotic response.

The use of ATC codes and Chemical Abstracts topics terms to associate molecule target sites for assessing model predictive performance is interesting but not sufficient for demonstrating the utility of the model for VS on phenotypic endpoints. That the molecules prioritized by the model are frequently associated with CNS therapeutics does not demonstrate that the model is useful for predicting the anti-phagocytotic response. The work would be much stronger with follow-up testing on some or all of the 64 prioritized molecules from SWEETLEAD in the original SPARK assay. Are there other molecules (outside of SWEETLEAD) that have been tested in this assay (or similar assay) that could be evaluated by the model retrospectively? Are there other molecules shown to be active in published assay data (ChEMBL or PubChem BioAssay) on cell-based or biochemical targets related to synaptic phagocytotic activity? Perhaps these data could be used in retrospective test.

Minor Issues

Table 1 – the degree units on the quartile values should be removed. Median (50%) is usually reported when reporting quantiles.

Line 113 – Table 3 does not show that nearly all models have better validation performance with complex validation procedure.

SI Table 9 -- # trees range probably should go much higher – I see much larger forests used with ECFP input representations (n_estimators) in Liu et al., 2019. https://pubs.acs.org/doi/10.1021/acs.jcim.8b00363).

Materials and Methods does not indicate code, libraries, or packages used for model implementations. This is necessary in cases where default hyperparameters are not indicted in text (e.g, in Random Forest what were the max features per tree, etc).

**Have the authors made all data and (if applicable) computational code underlying the findings in their manuscript fully available?**

Reviewer #1: **No: **link https://github.com/gravins/DGNs-for-schizophrenia is empty.

Reviewer #2: **No: **link to GitHub repo in manuscript but code/data not yet uploaded.

PLOS authors have the option to publish the peer review history of their article (what does this mean?). If published, this will include your full peer review and any attached files.

Reviewer #1: No

Reviewer #2: No
---

## [Decision Letter · Decision Letter 1]

29 Mar 2022

Dear Mr. Gravina,

We are pleased to inform you that your manuscript 'Controlling astrocyte-mediated synaptic pruning signals for schizophrenia drug repurposing with Deep Graph Networks' has been provisionally accepted for publication in PLOS Computational Biology.

Best regards,

Carl Herrmann, Ph.D.

Associate Editor

PLOS Computational Biology

Feilim Mac Gabhann

Editor-in-Chief

PLOS Computational Biology

Reviewer's Responses to Questions

**Comments to the Authors:**

Reviewer #1: The authors have addressed my concerns.

Reviewer #2: Authors have effectively addressed my concerns in previous review. Nice work--was a pleasure to read!

**Have the authors made all data and (if applicable) computational code underlying the findings in their manuscript fully available?**

Reviewer #1: Yes

Reviewer #2: Yes

PLOS authors have the option to publish the peer review history of their article (what does this mean?). If published, this will include your full peer review and any attached files.

Reviewer #1: No

Reviewer #2: No

---

## [Editor Report · Acceptance letter]

29 Apr 2022

PCOMPBIOL-D-21-01801R1 

Controlling astrocyte-mediated synaptic pruning signals for schizophrenia drug repurposing with Deep Graph Networks

Dear Dr Gravina,

I am pleased to inform you that your manuscript has been formally accepted for publication in PLOS Computational Biology. Your manuscript is now with our production department and you will be notified of the publication date in due course.

With kind regards,

Andrea Szabo
